# Relationship between Pharmacokinetic/Pharmacodynamic Target Attainment and Microbiological Outcome in Critically Ill COVID-19 Patients with Documented Gram-Negative Superinfections Treated with TDM-Guided Continuous-Infusion Meropenem

**DOI:** 10.3390/pharmaceutics14081585

**Published:** 2022-07-29

**Authors:** Maria Sanz Codina, Milo Gatti, Carla Troisi, Giacomo Fornaro, Zeno Pasquini, Filippo Trapani, Andrea Zanoni, Fabio Caramelli, Pierluigi Viale, Federico Pea

**Affiliations:** 1Department of Clinical Pharmacology, Medical University of Vienna, 1090 Vienna, Austria; maria.sanzcodina@meduniwien.ac.at; 2Department of Medical and Surgical Sciences, Alma Mater Studiorum University of Bologna, 40138 Bologna, Italy; carla.troisi2@unibo.it (C.T.); pierluigi.viale@unibo.it (P.V.); federico.pea@unibo.it (F.P.); 3Clinical Pharmacology Unit, Department for Integrated Infectious Risk Management, IRCCS Azienda Ospedaliero-Universitaria di Bologna, 40138 Bologna, Italy; 4Infectious Diseases Unit, Department for Integrated Infectious Risk Management, IRCCS Azienda Ospedaliero-Universitaria di Bologna, 40138 Bologna, Italy; giacomofornaro1990@gmail.com (G.F.); zeno.pasquini@gmail.com (Z.P.); filippofabio.trapani@aosp.bo.it (F.T.); 5Division of Anesthesiology, Department of Anesthesia and Intensive Care, IRCCS Azienda Ospedaliero-Universitaria di Bologna, 40138 Bologna, Italy; andrea.zanoni@aosp.bo.it; 6Pediatric Intensive Care Unit, IRCCS Azienda Ospedaliero-Universitaria di Bologna, 40138 Bologna, Italy; fabio.caramelli@aosp.bo.it

**Keywords:** critically ill patients, COVID-19, meropenem, PK/PD target attainment, microbiological failure, Gram-negative superinfections

## Abstract

Objectives: The objective of this study was to explore the relationship between pharmacokinetic/pharmacodynamic (PK/PD) target attainment of continuous-infusion (CI) meropenem and microbiological outcome in critical COVID-19 patients with documented Gram-negative superinfections. Methods: Patients receiving CI meropenem for documented Gram-negative infections at the COVID ICU of the IRCCS Azienda Ospedaliero-Universitaria di Bologna and undergoing therapeutic drug monitoring from January 2021 to February 2022 were retrospectively assessed. Average steady-state meropenem concentrations (C_ss_) were calculated and the C_ss_/MIC ratio was selected as a pharmacodynamic parameter of meropenem efficacy. The C_ss_/MIC ratio was defined as optimal if ≥4, quasi-optimal if between 1 and 4, and suboptimal if <1. The relationship between C_ss_/MIC and microbiological outcome was assessed. Results: Overall, 43 critical COVID-19 patients with documented Gram-negative infections were retrieved. Combination therapy was implemented in 26 cases. C_ss_/MIC ratios were optimal in 27 (62.8%), quasi-optimal in 7 (16.3%), and suboptimal in 9 cases (20.9%). Microbiological failure occurred in 21 patients (48.8%), with no difference between monotherapy and combination therapy (43.8% vs. 53.8%; *p* = 0.53). The microbiological failure rate was significantly lower in patients with an optimal C_ss_/MIC ratio compared to those with a quasi-optimal or suboptimal C_ss_/MIC ratio (33.3% vs. 75.0%; *p* = 0.01). Conclusion: Suboptimal attainment of meropenem PK/PD targets may be a major determinant impacting on microbiological failure in critical COVID-19 patients with Gram-negative superinfections.

## 1. Introduction

The SARS-CoV-2 pandemic has been responsible for most intensive care unit (ICU) admissions in the last two years, accounting for high morbidity and mortality [1]. Bacterial colonization and superinfections have been described in critically ill COVID-19 patients with an incidence ranging from 27% to 40% [2]. A remarkable proportion of these superinfections were caused by Gram-negative pathogens, including multidrug-resistant (MDR) *Enterobacterales* and non-fermenting isolates, leading to a significant rise in antibiotic consumption in the ICU setting [3,4,5].

Although the novel approved beta-lactams have significantly enhanced the available therapeutic options against MDR Gram-negative pathogens [6], meropenem still remains the first-line therapy for the management of ICU patients affected by extended-spectrum beta-lactamase (ESBL)-producing *Enterobacterales*, and a valuable alternative for non-fermenting isolates exhibiting a permissive minimum inhibitory concentration (MIC) [7,8].

The minimum pharmacodynamic (PD) target of efficacy for meropenem is considered a time of 40% of the dosing interval during which the plasma concentrations exceed the pathogen MIC (40%*t*
_> MIC_) [9]. However, recent evidence proposed the achievement of aggressive pharmacokinetic/pharmacodynamic (PK/PD) targets (namely, up to 100%*t*
_> 4–8×MIC_) in order to maximize clinical efficacy and minimize resistance development in critically ill patients [10,11,12].

Considering that sepsis-related pathophysiological alterations may significantly affect both the volume of distribution and the clearance of meropenem in critically septic patients, achieving the optimal PK/PD target may be challenging [13]. In this scenario, administering meropenem by continuous infusion (CI) may maximize the achievement of aggressive PK/PD targets while avoiding undesirable fluctuations in serum concentrations and preventing high peak levels that could be potentially associated with toxicity. CI administration has been shown to be superior compared to intermittent infusion in attaining a given pharmacodynamic (PD) target of efficacy and in improving clinical outcomes with beta-lactams among critically ill patients [14,15]. However, real-world data assessing the attainment of the PK/PD target of CI meropenem in critically ill COVID-19 patients are currently limited.

The purpose of this study was to assess the relationship between PK/PD target attainment and microbiological outcome in a cohort of critically ill COVID-19 patients affected by documented Gram-negative superinfections, treated with CI meropenem.

## 2. Materials and Methods

All the critically ill COVID-19 patients who were treated with CI meropenem because of suspected or documented Gram-negative superinfections at the COVID ICU of the IRCCS Azienda Ospedaliero-Universitaria di Bologna from 1 January 2021 to 28 February 2022 were retrospectively retrieved. Inclusion criteria were: (1) use of CI meropenem for at least 72 h; (2) at least one meropenem therapeutic drug monitoring (TDM) performed during treatment; (3) isolation of Gram-negative pathogens from microbiological cultures and determination of meropenem susceptibility.

Meropenem was prescribed at the discretion of the treating physician or infectious disease consultant according to current clinical practice implemented at the IRCCS Azienda Ospedaliero-Universitaria di Bologna. Treatment with meropenem was always started with a loading dose (LD) of 2 g over a 2 h infusion followed by a maintenance dose (MD) administered by CI. Considering that meropenem (both brand and generic formulations) is stable in aqueous solution for no more than 8 h [16,17], in order to grant active moiety during CI, the total daily MD was divided into three or four doses that were reconstituted every 6–8 h and infused over 6–8 h. MD regimens were initially selected according to renal function and underlying pathophysiological conditions and subsequently optimized by means of TDM.

Blood samples were collected in the first 48–72 h from the beginning of the treatment to determine meropenem C_ss_. Meropenem total blood concentrations were measured at the hospital’s Unique Metropolitan Laboratory and analyzed by means of a commercially available liquid chromatography-tandem mass spectrometry (LC–MS/MS) method (Chromsystems Instruments & Chemicals GmbH, Munich, Germany) [18]. TDM results were made available via the intranet to the MD clinical pharmacologist who provided expert clinical pharmacological advice for prompt dosing adaptation by ICU physicians within 6 h of blood collection.

Demographic (age, sex, weight, body mass index (BMI)) and clinical/laboratory data (need for mechanical ventilation and vasopressors, implementation of continuous renal replacement therapy (CRRT) or extracorporeal membrane oxygenation (ECMO), creatinine clearance (CLCr), site/type of infections, bacterial clinical isolate, MIC for meropenem, average meropenem C_ss_, implementation of antibiotic combination therapy, microbiological and clinical outcome, ICU and 30-day mortality rates) were extracted for each patient. Combination therapy was defined as the concomitant use of other antibiotics active against Gram-negative pathogens (aminoglycosides, colistin, fosfomycin, fluoroquinolones, tigecycline, and cotrimoxazole).

All of the patients undergoing CRRT received continuous venovenous hemodiafiltration (CVVHDF) with regional citrate anticoagulation by a Prismaflex Gambro machine and Prismaflex set ST 150 (AN 69 ST membrane with a surface area of 1.5 m^2^). The blood flow rate was maintained between 100 and 150 mL/min. The predilution citrate solution flow rate was automatically set for ensuring a circuit blood citrate concentration of 2.5–3 mmol/L. The predilution infusion flow rate/dialysate flow rate ratio was fixed at 1:1. The postdilution solution flow was fixed at 300 mL/h. A dialysis dose of 30–35 mL/Kg/h of the total effluent volume was prescribed.

Documented bloodstream infection (BSI) was defined as the isolation of Gram-negative pathogens from blood cultures. Documented ventilator-associated pneumonia (VAP) was defined as the presence of a Gram-negative bacterial load ≥10^4^ CFU/mL in the bronchoalveolar lavage (BAL) fluid culture documented after more than 48 h of endotracheal intubation and initiation of mechanical ventilation [19]. Documented complicated urinary tract infection (cUTI) was defined as the presence of a Gram-negative bacterial load ≥10^5^ CFU/mL in the urine culture [20].

The MIC of the identified Gram-negative pathogens was determined by means of the semi-automated broth microdilution method (Microscan Beckman NMDRM1) for *Enterobacterales* and *Pseudomonas aeruginosa*, or by a broth microdilution panel (ITGN10) for *Acinetobacter baumannii*. MICs were interpreted according to the European Committee on Antimicrobial Susceptibility Testing (EUCAST) clinical breakpoints. The percentage of time during which meropenem concentrations were above the MIC was selected as the best pharmacodynamic parameter for describing efficacy in terms of microbiological outcome and was expressed as the C_ss_/MIC ratio according to CI administration. In patients having multiple Gram-negative isolates, the C_ss_/MIC ratio was calculated using the higher MIC value. The C_ss_/MIC ratio was defined as optimal if ≥4, quasi-optimal if between 1 and 4, and suboptimal if <1. Thresholds were selected according to preclinical models showing that a C_ss_/MIC ≥ 4 may be associated with suppression of emergence of resistance to β-lactams [21]. Microbiological failure was defined as the persistence of the same bacterial pathogen in blood, BAL, or urine cultures after ≥7 days from starting meropenem treatment, as previously reported [22]. Microbiological eradication was defined as the occurrence of negativity of blood, BAL, or urine cultures in at least one subsequent assessment.

Descriptive statistics were used. Continuous data were presented as the mean ± standard deviation (S.D.) or median and interquartile range (IQR), whereas categorial variables were expressed as the count and percentage. Univariate analysis was performed by applying the chi-square test or Fisher’s exact test as appropriate. This study was approved by the Ethics Committee of the IRCCS Azienda Ospedaliero-Universitaria di Bologna (n. 442/2021/Oss/AOUBo approved on 28 June 2021).

## 3. Results

In the study period, a total of 212 critically ill patients who underwent TDM-guided meropenem in the IRCSS Azienda Ospedaliero-Universitaria di Bologna were screened. Among them, 67 were admitted to the COVID ICU owing to acute respiratory distress syndrome caused by severe COVID-19 pneumonia. Finally, forty-three patients had documented Gram-negative superinfections and were included in the study (Figure 1).

The demographics and clinical features of the included patients are reported in Table 1.

The mean ± S.D. age was 59.0 ± 13.8 years with a slight male preponderance (60.5%). The mean ± S.D. BMI was 30.7 ± 14.9 kg/m^2^. All patients required mechanical ventilation. Most patients (81.4%) had septic shock. Seventeen patients (39.5%) underwent CRRT and eight (18.6%) required ECMO. The overall ICU and 30-day mortality rates were 46.5%. The median (IQR) SOFA score was 9 (7–11) points.

The types of infection were VAP (38/43; 88.4%), BSI (13/43; 30.2%), and cUTI (4/43; 9.3%). In eight patients (18.6%), BSI and VAP occurred simultaneously. In two cases, concomitant VAP and cUTI were documented, while in one patient BSI, VAP, and cUTI occurred simultaneously. Overall, 47 Gram-negative pathogens were isolated, with *Pseudomonas aeruginosa* (29.8%), *Acinetobacter baumannii* (21.3%), and *Klebsiella pneumoniae* (12.8%) being the most frequent. Enterobacterales accounted for 48.9% of the overall Gram-negative isolates.

Combination therapy was adopted in 26 out of 43 cases (60.5%). Fosfomycin (15/26; 57.8%) and colistin (7/26; 27.0%) were the concomitant agents most frequently used, followed by one case each for ceftazidime-avibactam, ciprofloxacin, cotrimoxazole, and colistin + fosfomycin (Table 1). A total of 138 TDMs of meropenem were determined, with a median (IQR) of 3 (2–4) per patient. The median (IQR) meropenem C_ss_ average was 22.4 mg/L (12.3–32.5 mg/L). Overall, meropenem dosing adjustments were recommended in 51 out of 138 TDM assessments (37.0%, divided into 25.4% decreases and 11.6% increases). The starting meropenem dosing regimen was adjusted after the first TDM assessment in 17 out of 43 patients (39.5%, divided into 30.2% decreases and 9.3% increases). Among patients undergoing CVVHDF, meropenem dosing adjustments at the first TDM assessment were recommended in 10 out of 17 cases (58.8%, divided into 41.2% decreases and 17.6% increases). The C_ss_/MIC ratio was optimal in 27 cases (62.8%), quasi-optimal in 7 cases (16.3%), and suboptimal in 9 cases (20.9%).

Microbiological failure occurred in 21 patients (48.8%) and concerned 17 VAP, 2 BSI, and 2 VAP plus BSI cases. Overall, the microbiological failure rate was significantly lower in patients with an optimal C_ss_/MIC ratio compared to those with a quasi-optimal or suboptimal C_ss_/MIC ratio (33.3% vs. 75.0%; *p* = 0.01). No difference in microbiological failure emerged between patients treated with monotherapy and those receiving meropenem in combination therapy (43.8% vs. 53.8%; *p* = 0.53). Among the 17 patients who received meropenem monotherapy, the microbiological failure rate was significantly higher in those with a suboptimal C_ss_/MIC ratio compared to those with an optimal or a quasi-optimal C_ss_/MIC ratio (100.0% vs. 28.6%, *p* = 0.05; Figure 2a).

Among the 26 patients treated with meropenem combination therapy, a trend toward a higher microbiological failure rate occurred in those with a suboptimal C_ss_/MIC ratio compared to those with an optimal or a quasi-optimal C_ss_/MIC ratio (83.3% vs. 45.0%, *p* = 0.16; Figure 2b).

The microbiological failure rates among patients with VAP compared to those with BSIs were higher both in the monotherapy (53.3% vs. 33.3%) and combination therapy groups (52.2% vs. 28.6%). In regard to Gram-negative pathogens, the microbiological failure rate was significantly higher in patients with infections caused by non-fermenting Gram-negative pathogens (66.7%) compared to those with infections caused by *Enterobacterales* (21.7%, *p* = 0.002; Figure 3).

## 4. Discussion

Our study assessed the relationship between the PK/PD target attainment of CI meropenem and microbiological outcome in the novel scenario of critically ill COVID-19 patients with documented Gram-negative superinfections. The findings suggest the remarkable role that the achievement of an optimal and/or a quasi-optimal PK/PD target by means of a real-time TDM-guided approach may play in enabling microbiological cure with CI meropenem.

Critically ill COVID-19 patients are at high risk of bacterial superinfections, which in most cases are caused by MDR Gram-negative pathogens [2,3]. In this setting, CI meropenem may be a valuable therapeutic strategy for the management of infections caused by ESBL-producing *Enterobacterales* and/or non-fermenting pathogens, either as a monotherapy or combination therapy [8,24]. The achievement of C_ss_ > 4–5-fold above the MIC was shown to be helpful in minimizing the risk of microbiological failure and of carbapenem resistance development [12,21].

Notably, our findings show that suboptimal meropenem PK/PD target attainment accounted for most of the cases with microbiological failure. Furthermore, there was a trend toward a proportional increase in microbiological failure of meropenem therapy when PK/PD target attainment shifted from optimal to quasi-optimal and suboptimal, as previously reported with other traditional and novel beta-lactams [12,25]. Overall, these findings may support the utility of a TDM-guided approach in promptly assessing the achievement of the optimal meropenem PK/PD target in critically ill COVID-19 patients with documented Gram-negative superinfections.

It is noteworthy that no major difference in the microbiological failure rate was observed between patients receiving meropenem monotherapy and patients receiving combination therapy. Conversely, in both groups, there was a trend toward a proportional increase in the microbiological failure rate when PK/PD target attainment of meropenem shifted from optimal to suboptimal. These findings may offer further support to the contention that PK/PD target maximization of meropenem monotherapy may play a role in enabling microbiological cure that is more determinant than that played by the use of combination therapy. Indeed, combination therapy provided no benefits over monotherapy in terms of clinical and microbiological outcomes, as previously reported [26,27,28].

In our cohort, fosfomycin was the agent most frequently combined with meropenem. Although in vitro studies showed that fosfomycin may be synergic with meropenem against *Pseudomonas aeruginosa* [29], no real-world evidence currently supports the superiority of this combination therapy over monotherapy. This supports the idea that combination therapy with fosfomycin should not be pursued routinely, but should be reserved only when in the presence of difficult-to-treat strains [30,31].

It is noteworthy that most of the microbiological failures occurred in patients with VAP. This suggests that microbiological eradication may be especially difficult in deep-seated infections compared to bloodstream or urinary infections. This may be due to the limited penetration rate of meropenem into the epithelial lining fluid (ELF), which is approximatively 30% [32,33]. Overall, these findings may support the rationale for administering higher doses of CI meropenem in patients with VAP. This approach, by increasing lung exposure, may allow the achievement of aggressive PK/PD targets even at this difficult-to-treat infection site. In this regard, approaches focused on assessing the antibiotic concentration at the infection site could be helpful [34].

It was not unexpected that patients with infections caused by Gram-negative non-fermenting pathogens had a worse microbiological outcome compared to those having infections caused by *Enterobacterales* (most of which produce ESBLs). This could be related either to higher MICs for meropenem, which make the achievement of optimal meropenem PK/PD targets more challenging, or to the presence of multiple and complex mechanisms of resistance [34]. We could also speculate that the PD target of meropenem needed for eradicating non-fermenting pathogens could be higher than that needed for eradicating *Enterobacterales* strains according to preclinical data [35,36,37]. However, even in this challenging scenario, it appeared that maximization of PK/PD targets may be the major determinant of microbiological outcome, as achieving quasi-optimal PK/PD targets allowed the eradication of infections due to *Acinetobacter baumannii* or *Pseudomonas aeruginosa* even when in the presence of an MIC of up to 16–32 mg/L.

Our study has some limitations. The retrospective monocentric study design and the limited sample size of patients should be acknowledged. The assessment of bacterial susceptibility by means of the semi-automated broth microdilution method for *Enterobacterales* and *Pseudomonas aeruginosa* must be recognized. However, the good trend of the relationship between PK/PD target attainment and microbiological outcome is a major strength.

In conclusion, our findings suggest that achieving optimal PK/PD targets may be a major determinant of microbiological cure of Gram-negative superinfections treated with CI meropenem in critically ill COVID-19 patients. Higher PK/PD targets may be desirable in VAP sustained by non-fermenting Gram-negative pathogens for maximizing efficacy in this difficult-to-access site. Additionally, our findings highlight the potentially relevant role that a real-time TDM-guided strategy may play in this challenging scenario. Obviously, this study is simply a proof of concept, and large prospective clinical studies are warranted for confirming our hypothesis.

## Figures and Tables

**Figure 1 pharmaceutics-14-01585-f001:**
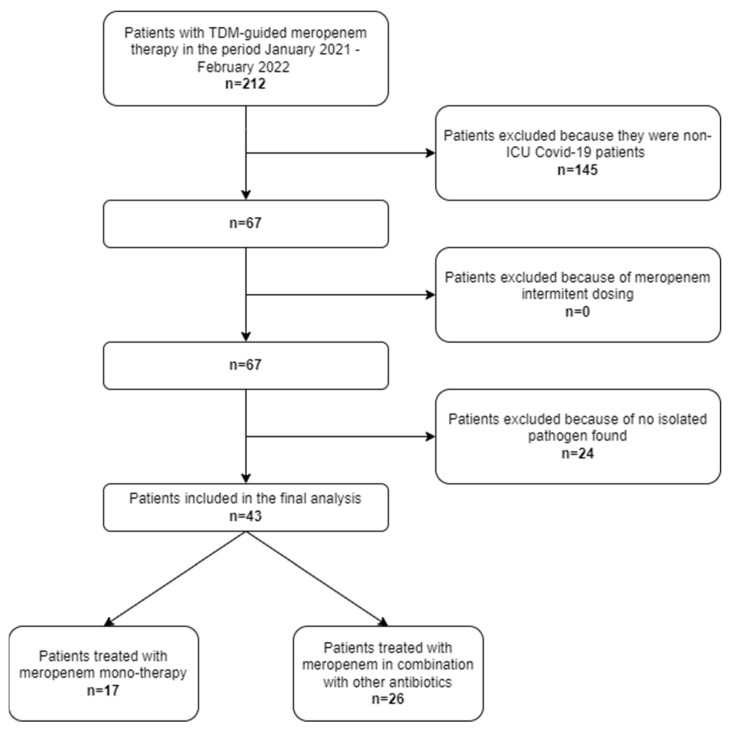
Flowchart of patient inclusion and exclusion criteria.

**Figure 2 pharmaceutics-14-01585-f002:**
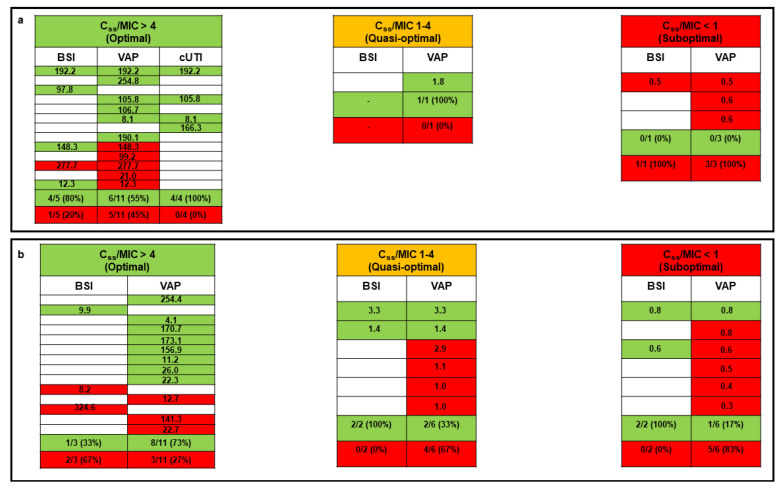
Relationship between pharmacokinetic/pharmacodynamic target attainment (expressed as the average C_ss_/MIC ratio) and microbiological outcome for critically ill COVID-19 patients treated with CI meropenem in monotherapy (**panel a**) or combination therapy (**panel b**). Green box, microbiological eradication; red box, microbiological failure; white box, absence of specific type of infection. Each row corresponds to a single patient. The C_ss_/MIC ratio is shown for each patient and defined as optimal if ≥4, quasi-optimal if between 1 and 4, and suboptimal if <1. BSI: bloodstream infection; C_ss_: meropenem average steady-state concentrations; cUTI: complicated urinary tract infection; MIC: minimum inhibitory concentration; VAP: ventilator-associated pneumonia.

**Figure 3 pharmaceutics-14-01585-f003:**
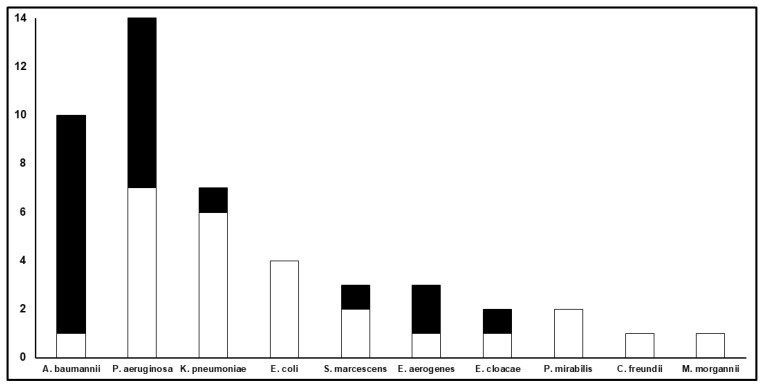
Microbiological outcome according to specific Gram-negative isolates. The *Y*-axis refers to the total number of clinical isolates for each Gram-negative pathogen described on the *X*-axis. The white part of each bar refers to the number of microbiological eradications, whereas the black part refers to the number of microbiological failures.

**Table 1 pharmaceutics-14-01585-t001:** Demographics and clinical characteristics of patients included in the PK/PD analysis classified according to the administration of monotherapy or combination therapy with meropenem.

	Overall (*n* = 43)	Monotherapy (*n* = 17)	Combination Therapy (*n* = 26)
Demographics			
Age	59.0 ± 13.8	63.1 ± 12.6	56.3 ± 13.7
Gender (m/f)	26/17 (60.5%/39.5%)	10/7 (58.8%/41.2%)	16/10 (61.5%/38.5%)
BMI	30.7 ± 14.9	29.8 ± 7.8	31.2 ± 17.8
Baseline eGFR	84.4 ± 39.3	72.6 ± 42.8	92.6 ± 36.7
Severity of infection			
SOFA score	9 (7–11)	8 (6–11)	9.5 (7–11.75)
Mechanical ventilation	43 (100.0%)	17 (100.0%)	26 (100.0%)
Vasopressors	35 (81.4%)	12 (70.5%)	23 (88.5%)
Continuous renal replacement therapy	17 (39.5%)	6 (35.3%)	11 (42.3%)
Extracorporeal membrane oxygenation	8 (18.6%)	2 (11.8%)	6 (23.1%)
Site of infection			
VAP	38 (88.4%)	15 (88.2%)	23 (88.5%)
BSI	13 (30.2%)	7 (41.2%)	6 (23.1%)
UTI	4 (23.5%)	4 (23.5%)	0 (0.0%)
Isolates (resistant)	47 (28)	21 (7)	26 (21)
*E. coli **	4 (0)	4 (0)	0 (0)
*K. pneumoniae **	7 (3)	5 (1)	2 (2)
*P. aeruginosa ***	14 (9)	5 (2)	9 (7)
*A. baumannii ****	10 (9)	2 (1)	8 (8)
*P. mirabilis **	2 (1)	2 (1)	0 (0)
*C. freundii **	1 (0)	0 (0)	1 (0)
*S. marcensens **	3 (1)	0 (0)	3 (1)
*E. cloacae **	2 (1)	2 (1)	0 (0)
*M. morgannii **	1 (1)	1 (1)	0 (0)
*E. aerogenes **	3 (3)	0 (0)	3 (3)
Outcome			
Microbiological cure	22 (51.2%)	10 (58.8%)	12 (46.2%)
ICU mortality	20 (46.5%)	8 (47.1%)	12 (46.2%)
30-day mortality	20 (46.5%)	8 (47.1%)	12 (46.2%)

* Resistant strains intended to be ESBL producers. ** Resistant strains intended to be MDR or XDR isolates according to the classification proposed by Magiorakos et al. [23]. *** Resistant strains intended to be carbapenem-resistant.

## Data Availability

Not applicable.

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
