# Peer review of "Relationship between Pharmacokinetic/Pharmacodynamic Target Attainment and Microbiological Outcome in Critically Ill COVID-19 Patients with Documented Gram-Negative Superinfections Treated with TDM-Guided Continuous-Infusion Meropenem"

_pharmaceutics, 2022, doi:10.3390/pharmaceutics14081585_

Round 1

Reviewer 1 Report

The paper describes the experience with CI meropenem to CoVid19 patients treated for serious Gram-negative infections in ICU. This is of interest for a broader audience in view of the paucity of data on CI of meropenem for such and other patients. The paper is well written and the data analysis well done, and the conclusions appropriate.

There are some details missing, which would strengthen the report and make it easier for future use of CI of meropenem:

1. How many serum concentration measurements were performed in each patient ?

2. How many times was the dose of meropenem changed according to these measurements, e.g., one assumes that on finding Ps. aeruginose or A. baumannii the dose was increased to achieve Css> 4 x MIC ? The same information is pertained for patients on renal replacemnet therapy - was the dose decreased ?

3. In how many of these septic chok patients was the original dose insufficient due to increase in Vd ?

4. Table 4, E. cloacae withour capital letter

5. Figure 3: Explain black and white bars

6. P. 14 last page in discussion, 3. paragraph, line 7: What does "aloe" mean ? Should it be "allow" ?

4. Could the authors be more specific on duration of CI treatment ?

Author Response

Manuscript ID: pharmaceutics-1828847 entitled “Relationship between pharmacokinetic/pharmacodynamic target attainment and microbiological outcome in critically ill COVID-19 patients with documented Gram-negative superinfections treated with TDM-guided continuous infusion meropenem” by Sanz Codina et al.

Dear Editor,

We would like to thank you for the opportunity to resubmit a revised version of this manuscript. We appreciated the reviewers’ constructive comments. All have been carefully considered and incorporated, where and whenever possible, in the revision.

As suggested by reviewer #2, we provided to revise our English in the revised version of the manuscript.

Our point-by-point responses are provided below.

Q= QUERY; A= ANSWER

Reviewer #1

The paper describes the experience with CI meropenem to CoVid19 patients treated for serious Gram-negative infections in ICU. This is of interest for a broader audience in view of the paucity of data on CI of meropenem for such and other patients. The paper is well written and the data analysis well done, and the conclusions appropriate.

We thank the reviewer for appreciating our manuscript.

There are some details missing, which would strengthen the report and make it easier for future use of CI of meropenem:

Q1. How many serum concentration measurements were performed in each patient?

A1. We thank the reviewer for this important comment. We performed a total of 138 meropenem TDM in our cohort, accounting for a median of 3 (IQR 2-4) serum concentration measurements per patient. We added this data in Results section (refer to Line 188-189).

Q2. How many times was the dose of meropenem changed according to these measurements, e.g., one assumes that on finding Ps. aeruginose or A. baumannii the dose was increased to achieve Css> 4 x MIC? The same information is pertained for patients on renal replacemnet therapy - was the dose decreased?

A2. We thank the reviewer for this important comment, allowing us to add this relevant information. Overall, meropenem dosing adjustments were recommended in 51 out of the 138 determinations (37.0%), being 25.4% decreases and 11.6% increases. In regard to patients requiring CRRT, meropenem dosing reduction was recommended in 7 out of 17 patients at first TDM assessment (41.2%), and overall in 18 out of 67 determinations performed during CRRT (26.9%). We added this data in Results section (refer to Line 190-196).

Q3. In how many of these septic shock patients was the original dose insufficient due to increase in Vd?

A3. We thank the reviewer for this comment. At first TDM assessment, meropenem dosing adjustments were recommended in 17 out of 43 included patients (39.5%), being increased in four cases (9.3%) and decreased in 13 cases (30.2%). We added this data in Results section (refer to Line 190-196).

Q4. Table 4, E. cloacae withour capital letter

A4. Thank you for this suggestion: we removed capital letter from E. cloacae.

Q5.Qqs Q5. Figure 3: Explain black and white bars

A5. Thank you for this suggestion: we explained the meaning of black and white bars in figure legend.

Q6. P. 14 last page in discussion, 3. paragraph, line 7: What does "aloe" mean? Should it be "allow"?

A6. We thank the reviewer for this comment, a typo occurred at this point. We replaced “aloe” with “allow” (refer to Line 278).

Q7. Could the authors be more specific on duration of CI treatment?

A7. We thank the reviewer for this comment, allowing us to better clarify this point. In all included patients, treatment with meropenem was started with a loading dose of 2g infused over 2h followed by a maintenance dose (selected according to renal function and pathophysiological characteristics of each patient) administered by continuous infusion. Continuous infusion was maintained during the whole treatment duration with meropenem. Considering that meropenem (both brand and generic formulation) is stable in aqueous solution for no more than 6-8 hours (we replaced reference no. 16 with two more appropriate references, namely doi:10.1097/FTD.0000000000000054 [reference no. 16] and doi:10.1177/0018578718779009 [reference no. 17], also according to comment No. 1 of reviewer #2), the total daily maintenance dose was divided in three or four administrations, each of which was reconstituted every 6-8 h and infused over 6-8 hours. We clarified better this important issue in Methods section (refer to Line 95-98).

Reviewer 2 Report

This article was a very interesting study, but the study design, i.e., method, results, discussion are unclear.

The authors have listed reference 16 for stability, but the design of this study seems inappropriate as the attached reference would have a titer of less than 90% with the current study design.  Because, essentially, the most important PK/PD parameter of MEPM is the evaluation of TAM, which is not mentioned at all in this study. Therefore, we are conducting TDM on the spot and looking at Css/MIC, but the establishment of TAM achievement is unclear, and we cannot determine the appropriateness of this study design. Most recent reports recommend prolong dosing times for carbapenems and do not mention any considerations.

・An assessment of severity, such as the SOFA score, is required. If the severity of the disease is not known, the pros and cons of combination therapy may not be clear. In addition, without a description of the conditions for setting up CRRT in different countries, it is difficult to evaluate different conditions. 

・Figures 2 and 3 seem inappropriate. There is no statistical analysis and Figure 3 is incomprehensible because it is not annotated.

Author Response

Manuscript ID: pharmaceutics-1828847 entitled “Relationship between pharmacokinetic/pharmacodynamic target attainment and microbiological outcome in critically ill COVID-19 patients with documented Gram-negative superinfections treated with TDM-guided continuous infusion meropenem” by Sanz Codina et al.

Dear Editor,

We would like to thank you for the opportunity to resubmit a revised version of this manuscript. We appreciated the reviewers’ constructive comments. All have been carefully considered and incorporated, where and whenever possible, in the revision.

As suggested by reviewer #2, we provided to revise our English in the revised version of the manuscript.

Our point-by-point responses are provided below.

Q= QUERY; A= ANSWER

Reviewer #2

This article was a very interesting study, but the study design, i.e., method, results, discussion are unclear.

We thank the reviewer for appreciating our paper. We clarified and addressed methodological issues in response to comment No. 1.

Q1. The authors have listed reference 16 for stability, but the design of this study seems inappropriate as the attached reference would have a titer of less than 90% with the current study design.  Because, essentially, the most important PK/PD parameter of MEPM is the evaluation of TAM, which is not mentioned at all in this study. Therefore, we are conducting TDM on the spot and looking at Css/MIC, but the establishment of TAM achievement is unclear, and we cannot determine the appropriateness of this study design. Most recent reports recommend prolong dosing times for carbapenems and do not mention any considerations.

A1. We thank the reviewer for this comment, allowing us to better clarify these important issues. We replaced reference no. 16 with two more appropriate and updated references, namely doi:10.1097/FTD.0000000000000054 (reference no. 16) and doi:10.1177/0018578718779009 (reference no. 17), showing that degradation of meropenem was both time and temperature dependent, and the aqueous solutions were stable for up to 7.4-8 hours in the temperature range between 25°C and 35°C. Consequently, our study design is consistent with those reported in updated literature.

We agree with the fact that the PK/PD index associated with optimal beta-lactam activity (including meropenem) is the time above MIC, and preclinical studies reported that the minimum PD target of efficacy for meropenem is a 40-70% of time of the dosing interval during which plasma concentrations exceed the pathogen MIC (40–70%fT>MIC). However, critically ill patients may benefit from more aggressive PK/PD target (i.e., 100%fT>4-8 x MIC) focused on either maximizing efficacy and microbiological eradication, or on minimizing the risk of resistance emergence, as reported in several studies and guidelines (refer to reference No. 10-12 and no. 21). These concepts are detailed in the Introduction and Discussion sections (refer to Line 63-68 and 244-246). It is noteworthy clarifying that Css/MIC (average steady-state drug concentration over MIC) and TAM (duration of time that the free drug concentration remains above the MIC) are linked by the same rationale that is simply transposed in different formula with the same meaning. We better clarify this issue in Methods section (refer to Line 137-140).

We agree with the fact that prolonged infusion represents the best therapeutic strategy for maximizing the achievement of optimal meropenem PK/PD targets and for suppressing resistance emergence, particularly in critically ill patients. This concept was mentioned and discussed in Lines 243-248.

Q2. An assessment of severity, such as the SOFA score, is required. If the severity of the disease is not known, the pros and cons of combination therapy may not be clear. In addition, without a description of the conditions for setting up CRRT in different countries, it is difficult to evaluate different conditions.

A2. We thank the reviewer for these important suggestions. We reported median SOFA score of patients receiving monotherapy and combination therapy in Table 1 and in the Results section (refer to Line 176-177). Additionally, we added some relevant information regarding CRRT settings in our cohort in the Methods section (refer to Line 118-125).

Q3. Figures 2 and 3 seem inappropriate. There is no statistical analysis and Figure 3 is incomprehensible because it is not annotated.

A3. We thank the reviewer for this comment. We performed univariate regression analysis for better explaining the results showed in Figure 2 and Figure 3, and we edited the Abstract (refer to Line 41-43) and the Results sections (refer to Line 199-229) accordingly. Additionally, we added an appropriate legend in Figure 3 for explaining in detail the meaning of black and white bars.

Round 2

Reviewer 1 Report

The autors have responded appropriately to all comments of the reviewers. I have no further commenst.

Reviewer 2 Report

It appears that the concerns I had have been raised have been remedied.

There is one item that I would like to see added.

Please add the TAM results to Figure 2.

Author Response

RESPONSE TO REVIEWERS

Manuscript ID: pharmaceutics-1828847 entitled “Relationship between pharmacokinetic/pharmacodynamic target attainment and microbiological outcome in critically ill COVID-19 patients with documented Gram-negative superinfections treated with TDM-guided continuous infusion meropenem” by Sanz Codina et al.

Dear Editor,

We would like to thank you for the opportunity to resubmit a new revised version of this manuscript. We appreciated the reviewers’ constructive comments. All have been carefully considered and incorporated, where and whenever possible, in the revision.

Our point-by-point responses are provided below.

Q= QUERY; A= ANSWER

Reviewer #2

It appears that the concerns I had have been raised have been remedied.

There is one item that I would like to see added.

Please add the TAM results to Figure 2.

We thank the reviewer for appreciating our revised version of the manuscript. As required, we added in Figure 2 the TAM results for each patient, expressed in terms of Css/MIC ratio.
